# Hsa-miR-34a-5p and hsa-miR-375 as Biomarkers for Monitoring the Effects of Drug Treatment for Migraine Pain in Children and Adolescents: A Pilot Study

**DOI:** 10.3390/jcm8070928

**Published:** 2019-06-27

**Authors:** Luca Gallelli, Erika Cione, Fancesco Peltrone, Serena Siviglia, Antonio Verano, Domenico Chirchiglia, Stefania Zampogna, Vincenzo Guidetti, Luca Sammartino, Angelo Montana, Maria Cristina Caroleo, Giovambattista De Sarro, Giulio Di Mizio

**Affiliations:** 1Department of Health Sciences, University of Magna Graecia, 88100 Catanzaro CZ, Italy; 2Department of Pharmacy, Health and Nutritional Sciences, Department of Excellence 2018-2022, University of Calabria, 87036 Arcavacata, Rende CS, Italy; 3Operative Unit of Pediatric diseases, Pugliese Ciaccio Hospital, 88100 Catanzaro CZ, Italy; 4Department of Neurosurgery, University of Catanzaro, Campus Germaneto, 88100 Catanzaro CZ, Italy; 5Section of Child and Adolescent Neuropsychiatry, Department of Human Neuroscience, "Sapienza" University, 00185, Rome RM, Italy; 6Dentasam, 95128 Catania CT, Italy; 7Department of Medical Science, Surgical Science and advanced Technologies “G.F, Ingrassia”, University of Catania, 95124 Catania CT, Italy; 8Forensic Medicine, Department of Law, Economy and Sociology, Magna Graecia University of Catanzaro, 88100 Catanzaro CZ, Italy

**Keywords:** biomarkers, miRs, saliva, blood, pain-migraine, drug treatment, clinical risk management

## Abstract

MicroRNAs (miRs) have emerged as biomarkers of migraine disease in both adults and children. In this study we evaluated the expression of hsa-miR-34a-5p and hsa-miR-375 in serum and saliva of young subjects (age 11 ± 3.467 years) with migraine without aura (MWA), while some underwent pharmacological treatment, and healthy young subjects were used as controls. miRs were determined using the qRT-PCR method, and gene targets of hsa-miR-34a-5p and hsa-miR-375 linked to pain-migraine were found by in silico analysis. qRT-PCR revealed comparable levels of hsa-miRs in both blood and saliva. Higher expression of hsa-miR-34a-5p and hsa-miR-375 was detected in saliva of untreated MWAs compared to healthy subjects (hsa-miR-34a-5p: *p* < 0.05; hsa-miR-375 *p* < 0.01). Furthermore, in MWA treated subjects, a significant decrease of hsa-miR-34a-5p and of hsa-miR-375 was documented in saliva and blood compared to MWA untreated ones. Altogether, these findings suggested thathsa-miR-34a-5p and hsa-miR-375 are expressed equally in blood and saliva and that they could be a useful biomarker of disease and of drug efficacy in patients with MWA.

## 1. Introduction

Migraine is a common disabling primary headache disorder in both adult and children [1]. NSAIDs represent the first line treatment for acute migraine attack [2,3,4], but their use can induce chronic migraines and adverse drug reactions (ADRs), particularly in children or adolescents [5,6].

Identification of specific biomarkers of disease and of drug efficacy in children and adolescents with migraines have clinical, forensic and ethical issues. In spite of this concern, some authors have proposed the use of biomarkers to guide the choice of treatment, to monitor the improvement or worsening of migraine symptoms during the treatment and to determine the appropriate therapeutic regimen in order to avoid drug/dietary supplements ADRs or inefficacy [7,8,9]. Currently, a promising biomolecule appears to be Plasma Calcitonin Gene-Related Peptide (CGRP) [9], since its serum level increases during acute migraine attack [10] and decreases upon treatment with onabotulinumtoxin type A [11]. Moreover, Fan et al. [12] recently reported in a clinical trial performed on 68 patients with migraine, 30 with migraine-headache, and 22 free of migraine-headache, that plasma CGRP level could be able to differentiate migraine from non-migraine headache. However, other authors reported that it is possible to detect high CGRP levels only in cerebral vessel and ipsilateral jugular vein levels, but not in plasma [13,14,15]. Moreover, CGRP has a short half-life (about 7 min) [16,17] and could be liable for false negative results.

A cornerstone in modern medicine is undoubtedly the discovery of the family of non-coding RNA, also known as microRNA (miR). These biomolecules suppress target mRNA translation and stability [18] for a large segment of the transcriptome, and their influence on gene expression occurs within the cells producing them, as well as in remote cells through extracellular trafficking. Moreover, miR mediate also epigenetic modification in chronic neuropathic pain [19,20] and in migraine [21,22] and represent a functional interplay between serum miRs levels and drugs exposure [23] therefore they could play a role as biomarker candidate [24].

Moreover, in a clinical study, Andersen and colleagues [25] described an increased expression of has-miR-34a-5p in patients with migraine attacks, while Chen et al. [26] reported that hsa-miR-375 plays pivotal roles in cancer, metabolic, immune disease and inflammatory diseases.

Based on those observations, in this study we evaluated the expression of hsa-miR-34a-5p and hsa-miR-375 in serum and saliva of treated and untreated children with migraine without aura.

## 2. Materials and Methods

### 2.1. Study Subjects

We performed a prospective single center, parallel-group study from November 2017 to February 2019 in patients admitted to the Headache Unit of the Pediatric Operative Unit of the “Pugliese Ciaccio” Hospital of Catanzaro. The study protocol was approved by the Local Ethics Committee (Catanzaro Centro protocol number 233/2017), and the work was conducted in compliance with the Institutional Review Board/Human Subjects Research Committee requirements and with the Declaration of Helsinki and the Guidelines for Good Clinical Practice criteria. Before the beginning of the study, the signed informed consent was obtained from their parents.

### 2.2. Inclusion and Exclusion Criteria

Subjects with migraine without aura, of both sexes and aged from 4–17 years, with at least 4 attacks per month, were eligible to be enrolled. Migraine was diagnosed according to the criteria for pediatric age of the International Classification of Headache Disorders [27].

To eliminate confounding factors, subjects were excluded if they: Were aged <4 years or >18 years; obesity (body mass index >95 percentiles), or had a history of cardiovascular disease, hypertension, diabetes, hyperlipidemia, or stroke; a heavy smoker (>10 cigarettes/day); current pregnancy or lactation; infectious or inflammatory disease within a month prior to study commencement; had a major psychiatric disorder, such as depression or bipolar disorders according to DSM-5; or had a history of substance abuse, mental retardation (intelligence quotient <70), genetic syndromes (e.g., Down syndrome, Prader–Willi syndrome, fragile X syndrome), hypothyroidism, neuromuscular disorders, epilepsy, liver or renal diseases, gastrointestinal disorders such as peptic or duodenal ulcer, dyspepsia, or heartburn and hypersensitivity to NSAIDs. Patients enrolled were in migraine ictal phase.

### 2.3. Experimental Protocol

In a balanced recruitment study, subjects with migraine were enrolled in two groups:

Treated-group: subjects with migraine without aura that were on pharmacological treatment at the time of the observation (with magnesium or NSAIDs; i.e., acetaminophen).

Untreated-group: subjects with migraine without aura that presented to the observation for the first time and they did not receive any treatment.

Enrolled subjects (group 1 and 2) underwent a detailed clinical interview and a neurological examination during their first visit to the hospital in agreement with our previous papers [3,28]. Our questionnaires are designed to dig information about the following parameters: Demographic and clinical information; i.e., duration of migraine; drug use (for adolescents); headache frequency, duration, intensity (using visual analogue scale), location, quality, and triggers; and other symptoms such as: Unilateral pain; throbbing personality; aggravation by or avoidance of physical activity, moderate or severe in intensity; nausea or vomiting; and photophobia.

At the time of enrollment, samples of blood (5 mL) and saliva (2 mL) were collected in sterile tubes from both groups of migraine subjects.

In order to evaluate the expression of miRs respect to healthy subjects, 12 subjects age and sex matched with Group 2 were enrolled in this study as Control-Group.

### 2.4. Endpoints

The primary end-point was defined as a statistically significant difference (*p* < 0.05) in saliva or in blood expression of hsa-miR-34a-5p and hsa-miR-375 between Group 1 and 2. Another primary end-point was the absence of statistically significant differences (*p* < 0.05) in hsa-miR-34a-5p and/or hsa-miR-375 expression between serum and saliva in Group 1 and 2. Finally we also considered as a primary end point the statistically significant difference (*p* < 0.05) in the expression of hsa-microRNA-34a-5p and hsa-microRNA-375 in saliva between Group 2 and Control-Group.

### 2.5. MicroRNAs Extraction

Total RNA was extracted from 200 μL of biological fluid (blood and saliva) by using miRNeasy Serum/Plasma Kit (Qiagen, Venlo, The Netherlands), in order to lower possible contaminants previously described [29,30]. Briefly, samples were lysed with 5 volumes of QIAzol Lysis Reagent (Qiagen, Venlo, The Netherlands) for 5 min at room temperature and 200 μL of chloroform was added. The mixture was vortexed, kept at room temperature for 5 minutes, centrifuged at 12,000× *g* for 15 min at 4 °C and the upper (aqueous) phase was collected. Subsequently, 1.5 volumes of 100% ethanol were added to the aqueous phase. The solution obtained was passed through a RNeasy MinElute spin column in sequential 700 μL aliquots, where the total RNA binds to the membrane and phenol and other contaminants were efficiently washed away using specific buffers. Finally, RNA was washed once with 80% ethanol, dried for 2 min by centrifugation, and dissolved in 15 μL of RNase-free water.

### 2.6. MicroRNAs Loop Primer Method

The quantification of miR was performed with a method termed looped primer RT-PCR, following the Thermo Fisher Scientific protocol. Initially, 10 ng of total RNA was subjected to reverse transcription polymerase chain reaction using the TaqMan MicroRNA Reverse Transcription kit (Thermo Fisher Scientific, Waltham, MA, USA) for both, miRNA targets and endogenous control, according to manufacturer’s protocol. The thermocycling conditions were: 30 min at 16 °C, followed by 30 min at 42 °C, 5 min at 85 °C and 5 min at 4 °C.

### 2.7. Quantitative Real Time PCR (qRT-PCR)

The quantitative real-time polymerase chain reaction (qRT-PCR) was performed using TaqMan Universal PCR Master Mix Kit (Thermo Fisher Scientific), according to the manufacturer’s protocol and the equipment QuantumStudio3^TM^ Real-Time PCR Systems (Thermo Fisher Scientific, Waltham, MA, USA) miRNA quantitative PCR results were normalized to U6 as control levels. The reactions were performed in triplicate for each patient and incubated in optical 96-well reaction plates. The thermocycling conditions were: 95 °C for 10 min, and 40 cycles of 15 s at 95 °C, followed by 1 min at 60 °C.

After finalization of the qRT-PCR experiments, the average values of the cycle threshold (Ct) of the reactions in triplicate were determined. The comparative Ct method was adopted, and U6 was used as endogenous control. The relative expression of miRNAs tested were normalized by the average values of CtU6 and the Ct target miR–CtU6. The difference was plotted as 1/Δct × 10 directly.

### 2.8. In silicoPredictionof hsa-miR-34a-5p and hsa-miR-375 Target Genes

In order to identify genes as target of hsa-miRs testedwe performed in silico analysis. The in silico identification of the target genes was performed using miRWalk (http://zmf.umm.uni-heidelberg.de/apps/zmf/mirwalk2/index.html), miRDB (http://mirdb.org/), and DIANA-TOOLS human databases. Besides, String database (http://diana.imis.athena-innovation.gr/DianaTools/index) was used to check hsa-miRNAs’ target genes had already been validated experimentally for both hsa-miR-34a and hsa-miR-375.

### 2.9. Statistical Analysis

All data were analyzed using the IBM SPSS21 statistical program (Microsoft, Redmond, WA, USA) by evaluating the arithmetic characteristics, such as mean, geometric mean, standard deviation (SD), etc. Data parameters were checked for normality using the Shapiro-Wilk normality test and analyzed accordingly. The statistical analysis method for microRNAs was performed using unpaired *t*-test with Welch’s correction and Graph Pad 5.0 software (GraphPad Software, San Diego, CA, USA). The differences were considered significant for values of *p* < 0.05. Power calculation: this was a pilot study; therefore, we did not perform the power calculation. However, assuming a difference in proposed endpoint from baseline to end of treatment between each group of 0.5, with an SD of 0.80, approximately 33 patients per treatment group would be required to detect superiority of a group to the other groups with 80% power and a significance level of 5% for a two-sided test. Assuming 5% of patients would not meet the criteria for the full analysis set (FAS), ~104 randomized patients would be required for this study.

Considering that our study was a pilot study we enrolled 36 patients (12 in each group) and we had a power di 0.4 (the Type I error probability associated with this test of this null hypothesis is 0.05).

## 3. Results

### 3.1. Population

During the study, we enrolled 78 subjects with migraine (41 females and 38 males), age 13 ± 2.9 years. After clinical evaluation and laboratory investigations, 35 migraine subjects (19 females and 16 males) fulfilled the inclusion criteria, but only 24 of these (12 females and 12 males, age 11 ± 3.467 years) were included and completed the study. The remaining 11 subjects (5 females and 6 males) were excluded because the parents did not sign the consent (parents refused to take the blood samples). All enrolled subjects (*n* = 24) presented migraine without aura and were enclosed in the two groups:

Treated-group: *n* = 12 migraine subjects, 6 males and 6 females, between 7 and 15 years (mean 11 ± 2.976 years), chronically treated with magnesium (400 mg/day for 3 months) + acetaminophen (15 mg/kg) or ibuprofen (10 mg/kg) during the acute attack (Table 1).

Untreated-group: *n* = 12 migraine subjects, 6 males and 6 females, between 4 and 16 years (mean: 11 ± 3.750 years) (Table 1).

History showed that 13 of 24 subjects (54.2%) had a positive familiality with headache and this was more common in females (83.3%); moreover, patients enrolled in the treated-group had mild-moderate pain visual analogue scale (VAS < 5) while subjects in untreated-group had severe pain (VAS > 8) (Table 1). Any subjects presented comorbidity or were allergic to drugs or other substances and none of the enrolled subjects presented any adverse drug reactions during the study period.

Finally, 12 healthy subjects (6 males and 6 female) between 4 and 16 years (mean 11 ± 4.5 years), without migraine pain, were enrolled as control-group (Table 1).

### 3.2. Saliva Mirrors Serum Levels of hsa-miR-34a-5p and hsa-miR-375

Total RNA was extracted from saliva (*n* = 24) and serum frozen fractions (*n* = 24) and the percentage of the miRNAs recovery rate was about 70–75% for all samples analyzed. As shown in Figure 1A,B, saliva and serum frozen fractions obtained from all enrolled subjects with migraine (treated and untreated) constitutively expressed both hsa-miR-34a-5p and hsa-miR-375, without difference respect to age or gender. The qRT-PCR revealed comparable levels of hsa-miRs tested in both blood and saliva (Figure 1A,B), suggesting that saliva mirrored the serum expression profile and can be used by itself, instead of blood. That could be important for children care. Obtaining saliva instead blood is generally accepted by them. Moreover, qRT-PCR revealed expression of hsa-miR-34a-5p and hsa-miR-375 in saliva of migraine subjects enrolled in untreated subjects respect to both biological fluid (hsa-miR-34a-5p: *p* < 0.05; hsa-miR-375 *p* < 0.01), without difference respect to age or gender.

### 3.3. Pharmacological Therapies Modulate miRs Expression in Biological Fluids

Using qRT-PCR analysis, we detected a significant decrease of about 50% for hsa-miR-34a-5p in both serum (Figure 2A) and saliva (Figure 2B) of treated patients (Group 1) compared to untreated patients (Group 2) (*p* < 0.01), without difference respect to age or gender. Similarly, in both serum and saliva, qRT-PCR analysis revealed a significant decrease (about 50%) of hsa-miR-375 levels in treated patients (Group 1) compared to untreated ones (Group 2) (*p* < 0.05 and *p* < 0.01) (Figure 2C,D), without difference with respect to age or gender.

### 3.4. In silico Analysis of hsa-miR-34a-5p and hsa-miR-375 Target Gene

To predict the target genes of hsa-miR-34a-5p and hsa-miR-375 and identify possible genes involved in pain-migraine, we used 3 different tools. Eighty-eight predicted target genes were found for hsa-miR-34a-5p and thirty-nine were found for hsa-miR-375. Of these, genes linked to pain-migraine was chosen using DIANA-TOOLS for hsa-miR-34a-5p and hsa-miR-375 (Table 2 and Figure 3).

## 4. Discussion

Clinical assessment in children and adolescents suffering with pain-migraine is difficult, requiring the ability to build a trusting relationship with them but also with parents to achieve effective communication. Besides that, the ability of physicians to understand and evaluate pain-migraine related behaviors of children is challenge. Our study in children and adolescents, is the first that documented an increased expression of hsa-miR-34a-5p and hsa-miR-375 in both serum and saliva of untreated patients with migraine respect to both healthy people and treated migraine patients. Even if some authors reported that CGRP could represent biomarkers of migraine attack [9,10,12], its short half-life could be responsible of false negative results.

In modern medicine, miRs have emerged as novel diagnostic biomarkers in several diseases, including pain conditions, owing to their presence in virtually all biological fluids, resistance to enzymatic degradation [24,31] and detection at minute quantity. More recently, miRNAs’ deregulation has been documented in patients with migraine both during attacks and pain-free periods [32], highlighting the significance of these biomolecules in the pathophysiology of migraine and their potential use as potential biomarkers. However, these studies were performed in adults, while in our study we evaluated the miRs expression in children and adolescents. Our results demonstrated appreciable levels of hsa- miR-34a-5p and hsa-miR-375 in serum of children and adolescents with migraine without aura suggesting their involvement in the pathogenetic mechanisms of migraine. Moreover, we did not document any difference in miR expression with respect to the age and gender, documenting that these miRs are preserved and are not related to the sexual hormone’s activity. Previously, it has been demonstrated that an association between hsa-miR-34a-5p, inflammation and vascular endothelial response to stress exists, noting the positive correlation with the levels of proinflammatory cytokine IL-1β [33]. Notably a recent study [25] showed that acute migraine attacks in adults are associated with up-regulation of serum hsa-miR-34a-5p and hsa-miR-382-5p expression. In silico analysis performed with five different databases found both hsa-miR-34a-5p and hsa-miR-375 target several genes involved in pain transmission and soluble factors homeostasis suggesting a fine tuning of pain axis from these biomolecules more than a simple correlation with soluble inflammatory mediators. Andersen and colleagues find out that hsa-miR-34a-5p negatively modulated the expression gene has predicted target genes that facilitate GABAergic signaling. These include GABAA and GABAB receptor subunits and the ion-dependent GABA transporter SLC6A1suggesting a facilitator transmission promoting nociceptive effect. In addition, increased has-miR-34a-5p was recently found within dopaminergic circuits [34]. Furthermore, it has been shown that hsa-miR-375 binds the gene transcript encoding myotrophin, which regulates exocytosis and hormone release [35,36] suggesting that has-miR-375 may be involved in the secretion of neurotransmitters and in the regulation of transduction pathways involved in pain mechanisms. Finally, we documented a significant decreased expression of both miRs in both serum and saliva of patients under drug treatment (acute NSAIDs + chronic magnesium), with respect to untreated subjects, indicating a possible role of hsa-miR-34a-5p and hsa-miR-375 as biomarkers for the prediction of therapeutic response. This hypothesis is further strengthened by previous findings highlighting a functional cross-talk between pharmacological treatment and miRs, as drug administration can indirectly affect miR expression profiles [37], and conversely some specific miRs could be relevant as indicators of drug response [38].

Moreover, we documented higher expression of hsa-miR-34a-5p and hsa-miR-375 in saliva from children and adolescents with migraine without aura respect to healthy subjects, suggesting that miRs released into this peripheral fluid work as indirect readout of pathological processes arising into central nervous system since migraine is known to be associated with activation and sensitization of the trigeminal pain pathways [39,40]. In this view gene such HCN3, NV3, and GPR158were found in DIANA-TOOLS database associated in trigeminal neuralgia and targets of hsa-miR-34a-5p. In particular, HCN3 belong to hyperpolarization-activated potassium channels family proteins involved in trigeminal ganglion neuron activation [41], while NAV3 and GPR158 [42] are emerging in migraine.

Either in the miR walk database, we found that hsa-miR-375 targets gene involved in the vesicular trafficking and peptides/hormone release such as MTPN. Therefore, there is an interplay between those miRs that play a role in the trigeminal pain circuit. How this interplay occurs is still undefined, however, two potential routes are worth noting. First, the brain stem provides a potential CNS-to-oral cavity route via the sensory (V, VII, IX) and motor (XII, X, XII) nerves that innervate the salivary glands and tongue. Otherwise miRs delivery to the mouth could involve slow transport via the glymphatic system [43]. Notably, the expression of both hsa-miR-34a-5p and hsa-miR-375 were negatively modulated in saliva from children and adolescent underwent to treatment with NSAIDs and magnesium.

Monitoring disease and drug efficacy by dosing biomarkers on saliva samples from young subjects with migraine without aura could represent a clinical tool as mentioned before, but also a forensic tool in cases lacking objective assessment tools.

Appropriateness of patient monitoring and clinical and therapeutic choices could also be demonstrated by recording objective laboratory data as high quality standards to improve the quality of medical care, as suggested by clinical risk management.

This study has some major limitations represented by the limited sample size (particularly in control group), so we defined this study as a pilot study and other clinical trials in a large population must be performed to confirm these data.

In conclusion these preliminary results suggest that hsa-miR-34a-5p and hsa-miR-375 could be considered a biomarker of disease and of drug efficacy in young subjects with migraine without aura and that saliva could be used to monitor these biomarkers.

It is worth it to remind researchers that reliable tools assessment, reproducibility and objectivity are imperative in good clinical practice, and need to be consistent with the aim of improving quality of healthcare services, as provided by Italian Law 24/2017 [44]. This law is focused on professional liability and safety of care, based on guidelines and recommendations defined by scientific evidence, with the aim of improving good clinical healthcare practices and reducing litigation.

## Figures and Tables

**Figure 1 jcm-08-00928-f001:**
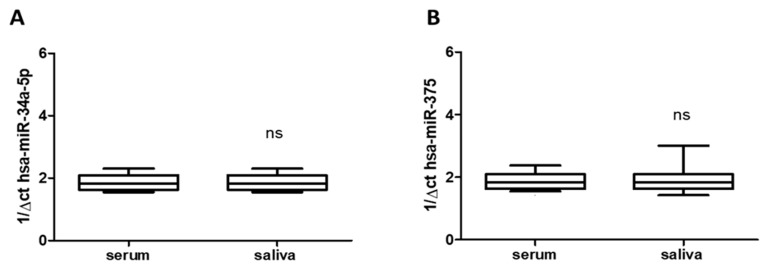
Expression of has-miR-34a-5p and has-miR-375 in serum (**A**) and in saliva (**B**) of migraines subjects showing not significant (ns) difference between them. miR data are plotted as box and whiskers 10:90 percentiles.

**Figure 2 jcm-08-00928-f002:**
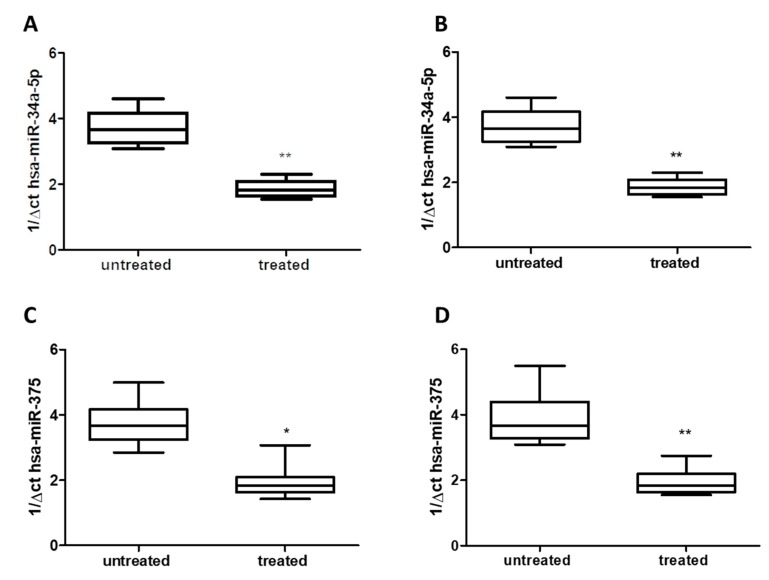
Expression of miRs in serum: (**A**) hsa-miR-34a-5p; (**C**) hsa-miR-375, and saliva: (**B**) hsa-miR-34a-5p, (**D**) hsa-miR-375: of either treated and untreated subjects with migraine. miRs data are plotted as box and whiskers 10:90 percentiles. * *p* < 0.05; ** *p* < 0.01.

**Figure 3 jcm-08-00928-f003:**
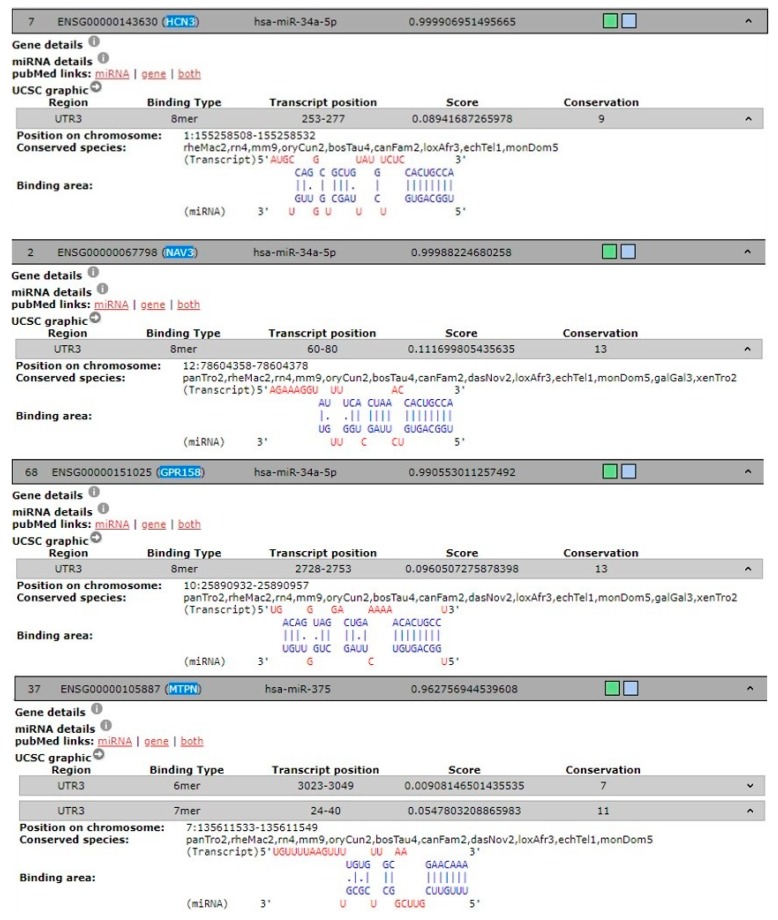
Predicted and experimentally validated interaction of hsa-miR-34a-5p and hsa-miR-375 (both in blue) with the corresponded 3’UTR mRNA genes (in red) from DIANA-TOOLS database.

**Table 1 jcm-08-00928-t001:** Clinical characteristics of treated and untreated groups of subjects with migraine without aura (MWA) and healthy group without MWA, herein named control, enrolled in our study. Patients enrolled in treated group (acetaminophen (15 mg/kg) or ibuprofen (10 mg/kg) during the acute attack) underwent also to supportive therapy with magnesium (400 mg/day for 3 months).

**Treated**					
**Sex**	**Age**	**Time of Pain**	**VAS**	**Familiality**	**Drugs**
Female	15	6 months	4	Father	Acetaminophen
Female	13	6 months	4	Sister	Ibuprofen
Female	12	6 months	3	NO	Acetaminophen
Female	12	6 months	4	Mother	Ibuprofen
Female	11	6 months	4	Mother	Acetaminophen
Female	9	6 months	3	NO	Ibuprofen
Male	15	6 months	3	Mother	Ibuprofen
Male	11	6 months	4	NO	Acetaminophen
Male	8	6 months	4	NO	Ibuprofen
Male	8	4 months	4	NO	Acetaminophen
Male	8	3 months	4	NO	Ibuprofen
Male	7	12 months	4	NO	Acetaminophen
**Untreated**					
**Sex**	**Age**	**Time of Pain**	**VAS**	**Familiality**	**Drugs**
Female	16	10 years	9	Mother	-
Female	14	6 months	8	Mother	-
Female	14	4 months	9	Father	-
Female	11	3 months	9	Mother	-
Female	8	12 months	8	Mother	-
Female	6	3 months	7	Mother	-
Male	16	12 months	9	NO	-
Male	13	3 months	8	NO	-
Male	12	12 months	9	Father	-
Male	12	6 months	9	NO	-
Male	7	5 months	8	Mother	-
Male	4	3 months	8	NO	-
**Control**					
**Sex**	**Age**	**Time of Pain**	**VAS**	**Familiality**	**Drugs**
Female	15	-	-	Father	-
Female	14	-	-	NO	-
Female	13	-	-	NO	-
Female	11	-	-	NO	-
Female	9	-	-	Mother	-
Female	6	-	-	Mother	-
Male	16	-	-	NO	-
Male	13	-	-	Mother	-
Male	12	-	-	NO	-
Male	12	-	-	NO	-
Male	7	-	-	NO	-
Male	4	-	-	NO	-

**Table 2 jcm-08-00928-t002:** Validated target genes linked to pain-migraine of hsa-miR-34a-5p according to the DIANA-TOOLS database. Validated target genes linked to exocytosis of hsa-miR-375 according to the DIANA-TOOLS database.

miRNA	Validated Target Genes	Associated Disease
**hsa-miR-34a-5p**	HCN3, NAV3, GPR158	trigeminal neuralgia
**hsa-miR-375**	MTPN	-

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
