# Peer review of "Hsa-miR-34a-5p and hsa-miR-375 as Biomarkers for Monitoring the Effects of Drug Treatment for Migraine Pain in Children and Adolescents: A Pilot Study"

_jcm, 2019, doi:10.3390/jcm8070928_

Reviewer 1 Report

The authors has studied "Hsa-miR-34a-5 and hsa-miR-375 as Biomarkers for Monitoring the Effects of Drug Treatment for Migrane Pain in Children and Adolescents. The strength of the article is that this has projected two important biomarkers for the migrane pain.
Major comments:
The main findings should be given at the Figure 1. Since many techniques has been used and most of such techniques should able to generate more orignial pictures from the experiment itself than showing on rather quantification.
Result does not seem too much for making points. I would recommended to focus more on the positive findings and try to elaborate it more details using different tools and technology to expand such findings.
Beside this, the article need to address below issues.

Line 22-25: Please use same font throughout the manuscript.

Line 25-27: Here, you may provide more generic type of abstracts.

Line 75: May be better subheading will be "Study Subject" or Just " Subject".

Line 83- 96: Both the paragraphs under "Inclusion Criteria" and "Exclusion criteria" Can be merged together with the single subheading "Inclusion and exclusion criteria"

Line 105-109: It may be good to rearrange these text in a Box 1 or Table 1 mentioning that "our questionnaires are design in such a way that it can  dig information about the below parameters" or any similar text.

Line 163: Maybe better to use "IBM SPSS 21" instead of "SPSS21"
Line 165: standard deviation short form should be (SD).
Line 166-167: Is it Student's t-test? If yes, you merge two sentences and rearrange better way to avoid redundancy of the word t-test twice here.
Line 176 : "di" what does it mean? Please explain which statistical tests are used?
Line 195: If "VAS" is used for the first time give the full form.
Line 211-213: Please come up wit first Figure that are with significant p-values or main findings.
line 221-222: Please provide actual experiment group name rather than "Group 1" or "Group 2"
Line 228: Since you have used five different tools here, can you highlight what  are the results did you obtain using such to strengthen your purpose of study ? Graphical representation of data or showing networks or their interaction may be useful or involvement of overlapping genes in Venn diagram could be better approach !
Line 316-422:References do not follow Journal's guideline, please correct it.      

Author Response

we thank the reviewer for these remarks. We revised our manuscript in agreement with your suggestions

Reviewer 2 Report

This study investigated the role of two miRNAs hsa-miR-34a-5p and hsa-miR-375 in migraine and its therapeutic effects in children and adolescents. The finding of this pilot study is interesting; however, several limitations impeded its scientific merits.

 Major:

1.      This study adopted a candidate target approach by specifically investigating two selected miRNAs. However, in my opinion, the reason for choosing these two miRNAs were not well justified. For example, it is reasonable to choose miR-34a-5p as a candidate as a previous study has demonstrated its potential role in migraine. However, the same study also found other miRNAs, especially miR-382-5p, to be important in migraine pathogenesis. There were also several miRNAs identified important for migraine in the other studies. Why the authors specifically chose miR-34a-5p but not the other miRNAs should be elaborated. Furthermore, the authors chose hsa-miR-375 as another candidate by saying that this miRNA plays pivotal roles in cancer, metabolic, immune disease and inflammatory diseases. There are many miRNAs having similar biological effects. However, it is unclear how the authors selected hsa-miR-375 out of a bunch of lists.

2.      As the authors had mentioned, sample size is a major limitation of this study, which makes this study severely underpowered. Diagnostic utility of these two miRNAs were not provided. In addition, there is no independent validation for this finding. The claim that these two miRNAs could be used as biomarkers could not be justified.

3.      More than half of the patients initially enrolled did not fulfill the inclusion criteria. The reason for their ineligibility should be provided.

4.      Headache frequency of both patient groups was not provided. Also, it is unclear whether these samples were collected in the interictal phase or ictal phase. Dynamics of migraine should be considered as previous studies have shown that some miRNAs could be upregulated during acute attacks.

5.      The results of in silico analysis were quite preliminary and could not provide sufficient insight to promote the understanding of migraine pathogenesis.

 Minor:

1.      In the abstract, the authors said that they also enrolled 12 patients as control group. Were these controls healthy people or patients?

2.      familiarity should be familiality

3.      It is seldom for a study to have multiple primary endpoints but no secondary endpoints.

4.      Group 2 have higher headache intensity than group 1. However, it is uncertain whether this headache intensity is pre-treatment baseline headache intensity or that after treatment.

5.      The authors may consider to shorten the speculative part of the discussion.

Author Response

we thank the reviewer for these remarks. we revised our manuscript in agreement with your suggestion

Round  2

Reviewer 1 Report

I still fell there is still much room for the manuscript to improve to reach high quality presentation. I would suggest not to mention Group 1, Group 2 or Group 3 in graph. But please provide actual experiments such as "Control" " Treated" "Untreated" to avoid diversion for reader (in fact, if you can provide actual pharmocological doses in index within Figure, could highly be more informative).

  Line 196-208: . Figure 1 is still lacking main findings that will be the first impression about the  whole manuscript. Right now you have non significant results in the first figure.

Earlier I have suggested to bring the main findings as the first figure to have a direction for the readers. But the author/s seem not to make that change ?

Line 22-37: In the abstract section, many naming for the sample size makes it more confusing. Could you write a generic abstract instead?. And if you want to give sample size, please provide 'n' together with p-values.

Line 26-27: 12 untreated subject and 12 healthy subjects: what is the differences between these two groups?

LIne 170 : What is power di?

Please provide actual dose of pharmacological treatment as you have mentioned in the graph and in the abstract as well.

Author Response

The authors have studied "Hsa-miR-34a-5 and hsa-miR-375 as Biomarkers for Monitoring the Effects of Drug Treatment for Migrane Pain in Children and Adolescents. The strength of the article is that this has projected two important biomarkers for the migrane pain.

We thank the reviewer for his/her general comment and for the concerns he/she raised. Absolutely these helped us to improve the overall quality of the manuscript. 

Major comments: 
The main findings should be given at the Figure 1. Since many techniques has been used and most of such techniques should able to generate more original pictures from the experiment itself than showing on rather quantification.
Result does not seem too much for making points. I would have recommended to focus more on the positive findings and try to elaborate it more details using different tools and technology to expand such findings. 
Beside this, the article need to address below issues.

Line 22-25: Please use same font throughout the manuscript. Done

Line 25-27: Here, you may provide more generic type of abstracts. 

Line 75: May be better subheading will be "Study Subject" or Just " Subject". Done

Line 83- 96: Both the paragraphs under "Inclusion Criteria" and "Exclusion criteria" Can be merged together with the single subheading "Inclusion and exclusion criteria". Done

Line 105-109: It may be good to rearrange these text in a Box 1 or Table 1 mentioning that "our questionnaires are design in such a way that it can dig information about the below parameters" or any similar text. Done

Line 163: Maybe better to use "IBM SPSS 21" instead of "SPSS21". Done
Line 165: standard deviation short form should be (SD).
Done
Line 166-167: Is it Student's t-test? If yes, you merge two sentences and rearrange better way to avoid redundancy of the word t-test twice here.
Done
Line 176 :
"di" what does it mean? Please explain which statistical tests are used?
Line 195: If "VAS" is used for the first time give the full form. 
Done
Line 211-213: Please come up with first Figure that are with significant p-values or main findings. 

Response: We thank the reviewer for this remark. Our aim with fig 1 is to prove that Saliva mirrors serum therefore there is not significant p-values between data.
line 221-222: Please provide actual experiment group name rather than "Group 1" or "Group 2"
Line 228: Since you have used five different tools here, can you highlight what  are the results did you obtain using such to strengthen your purpose of study ? Graphical representation of data or showing networks or their interaction may be useful or involvement of overlapping genes in Venn diagram could be better approach !

Response: We thank the reviewer for this very important remark/address. We redefined our analysis focusing on validated target gene in different database and we actually extended Table 2 with mRNAs-miRs interactions.
Line 316-422:References do not follow Journal's guideline, please correct it.

Reviewer 2 Report

The authors did not adequately address most of the issues raised previously. Specifically, they did not provide adequate rationale how and why these two miRNAs were chosen and why the others not. Literature review was also insufficient. Cases and controls were not well matched. Last but not least, the study is profoundly under-powered. 

Author Response

(The authors gave the same response as above.)
